# Oligomers of Carboxymethyl Cellulose for Postharvest Treatment of Fresh Produce: The Effect on Fresh-Cut Strawberry in Combination with Natural Active Agents

**DOI:** 10.3390/foods11081117

**Published:** 2022-04-13

**Authors:** Yevgenia Shebis, Elazar Fallik, Victor Rodov, Sai Sateesh Sagiri, Elena Poverenov

**Affiliations:** 1Agro-Nanotechnology and Advanced Materials Research Center, Department of Food Science, Agricultural Research Organization, The Volcani Institute, Rishon Lezion 7505101, Israel; yevgenia7777@gmail.com (Y.S.); biotech.satishsai@gmail.com (S.S.S.); 2The Robert H Smith Faculty of Agriculture, Food and Environment, The Hebrew University of Jerusalem, Rehovot 76100, Israel; 3Department of Postharvest Science, Agricultural Research Organization, The Volcani Institute, Rishon Lezion 7505101, Israel; efallik@volcani.agri.gov.il (E.F.); vrodov@volcani.agri.gov.il (V.R.)

**Keywords:** oligosaccharide, CMC, vanillin, food storability, fresh produce, strawberry

## Abstract

In this study, oligomers of carboxymethyl cellulose (O-CMC) were used as a new postharvest treatment for fresh produce. The oligomers were prepared by green and cost-effective enzymatic hydrolysis and applied to prevent spoilage and improve storability of fresh-cut strawberries. The produce quality was improved by all formulations containing O-CMC in comparison to the control, as indicated by the decrease in decay incidence, weight loss (min ~2–5 times less), higher firmness, microbial load decrease, better appearance, and sensorial quality of the fruits. Natural resources: ascorbic acid, gallic acid, and vanillin were further added to enhance the beneficial effect. O-CMC with vanillin was most efficient in all of the tested parameters, exhibiting the full prevention of fruit decay during all 7 days of refrigerated storage. In addition, fruits coated with O-CMC vanillin have the smallest weight loss (%), minimum browning, and highest antimicrobial effect preventing bacterial (~3 log, 2 log) and yeast/mold contaminations. Based on the obtained positive results, O-CMC may provide a new, safe, and effective tool for the postharvest treatment of fresh produce that can be used alone or in combination with other active agents.

## 1. Introduction

The ever-growing population across the globe demands a secure and safe food supply. This demand has been mainly addressed by designing and intensifying novel agriculture and postharvest treatment practices. However, approximately one-third of the total food produced for human consumption is getting wasted [1]. Ready-to-eat fresh-cut fruits and vegetables are one of the most perishable food categories characterized by high rate of losses and wastage [2]. To reduce the food loss, several approaches are being followed in postharvest processing of fresh produce [3]. One of the key challenges is to extend storage life and reduce the microbial spoilage of the produce without compromising its quality and safety [4]. For many years, the control of microbial contamination of fresh-cut produce was based on the extensive use of synthetic sanitizers such as chlorine. However, today these materials are banned in many European countries because of health concerns, and sustainable alternatives are sought by the industry [5]. In this regard, natural antimicrobial agents and protective edible coatings are of special interest [6,7,8]. Natural resources such as proteins, lipids, and polysaccharides have been employed to generate environmentally friendly, non-toxic, and cost-effective antimicrobial active packages and edible coatings to maintain and prolong the quality of produce. In addition, polysaccharides, pristine or in combination with other edible active agents, have been employed to generate effective coatings allowing to control spoilage, maintain texture and freshness, and keep the sensorial and nutritional quality of produce [9,10,11,12,13,14]. While the importance of polysaccharides is already widely acknowledged, the applied potential of oligosaccharides started attracting attention only in the recent period. The increased demand for novel and progressive postharvest methods has enabled the researchers to design and explore oligosaccharide-based treatments. Since oligosaccharides are short and linear structured sugar molecules, they facilitate the distinct design and allow to overcome polysaccharide shortcomings such as water solubility. Oligosaccharides can serve as prebiotics, plant elicitors, active drugs, anti-aging substances, feed additives to livestock, etc. [15,16,17,18]. Previous studies showed oligomer production from several common polysaccharides such as alginate, chitosan, and pectin. Their efficacy on postharvest produce was also tested. It was found that these oligomers with the incorporation of another active material such as organic acids and their salts (ascorbic/acetic/gallic acid), volatile compounds (aldehydes/terpenes/sulfur), and fatty acid esters (glyceryl monolaurate) minimize the decay percentage of fruit, preserving the fruit firmness, moisture, total soluble solid (TSS), and more [19,20,21,22,23]. One of the most common and widely used polysaccharides in the pharmaceutical, food, agriculture, and other industries, is carboxymethyl cellulose (CMC). CMC has been safely used as a food additive and is approved by the Food and Drug Administration (FDA) and European Food Safety Authority (EFSA) [24,25]. Although studies have reported about the degradation of carboxymethyl cellulose to oligomers [25,26,27], interestingly, these oligosaccharides are commercially unavailable and their activities are rather unexplored. To the best of our knowledge, applications of O-CMC for postharvest treatment have not been previously reported. 

In this work, we have produced O-CMC using an in-house developed, cost-effective and nonspecific enzymatic hydrolysis from widely available enzymes [28] and studied its effect on the postharvest quality of fresh-cut strawberries. The produced, novel O-CMC oligomers were applied alone and in combination with natural active agents, vanillin, gallic, and ascorbic acid, known to protect fruits and vegetables [29,30,31]. Since the implementation of oligosaccharides as a postharvest treatment is still budding, the development of a safe, green, water soluble, and cost-effective formulation on the basis of O-CMC is highly valued.

## 2. Materials and Methods

### 2.1. Materials

Pectinase from *Aspergillus aculeatus*, sodium acetate buffer solution (3 M), L-ascorbic acid (99%), vanillin (98%) ammonium bicarbonate, gallic acid, and absolute ethanol were purchased from Sigma-Aldrich/Merck KGaA (Darmstadt, Germany). Carboxymethyl cellulose sodium salt was supplied by Alfa Aesar (Heysham, UK). Deionized water was used for the preparation of all solutions.

### 2.2. Oligosaccharide Production

CMC oligosaccharides were produced by the non-specific enzymatic degradation of CMC according to the previously developed in-house procedure [22]. The CMC polysaccharide samples (0.5 g) were dissolved in 50 mL of sodium acetate buffer (50 mM, pH 5.2) with the addition of pectinase (130 µL, 3800 U/mL). The reaction mixture was incubated under the agitation of 230 rpm for 4 h at 47 °C. After the incubation, the reaction was stopped by placing the samples in a boiling water bath for 5 min, cooled down, and then brought to pH 7 using ammonium bicarbonate. The samples were then dissolved in ethanol at rising concentrations of 70–100% ethanol for the separation and purification of oligosaccharides, followed by centrifugation at 1200 rpm. Centrifugation was carried out multiple times until the entire residue was collected. Thus, the collected residue or precipitate was then freeze-dried to obtain the pure oligosaccharides as a powder. 

### 2.3. Application of Oligomers as Coatings on Fresh-Cut Fruit

#### 2.3.1. Fruit Preparation

Commercially picked strawberries (Aya cv.) were purchased at Akler farm, Kadima-Tzoran, HaSharon plain, Israel, and treated on the day of harvest immediately upon arrival. The fruit were washed with tap water containing 0.3 ppm active chlorine [5], rinsed with distilled water, and air-dried, with subsequent slicing of each fruit longitudinally into halves. To avoid size homogeneity (only big or small strawberries for each application), all were mixed and equally distributed for each test, resulting in heterogenic size distribution for each tested substance. 

#### 2.3.2. Treatment Preparation 

The Oligo-CMC solution (1% *w*/*v*) was prepared by dissolving 0.5 g of oligomers in 50 mL of distilled water at room temperature (25 °C) and stirred until full dissolution. Oligo-CMC solutions containing active substances (ascorbic acid/gallic acid/vanillin) were prepared in the ratio of 1:0.5 (*w*/*v*), by dissolving 0.5 g oligomers in 50 mL of distilled water with the addition of 0.25 g of the active substance and stirred until full dissolution. Controls for all active materials were prepared by dissolving 0.25 g of the chosen substance in 50 mL of distilled water.

#### 2.3.3. Treatment Application and Storage 

The trials included the following treatments: (a) water (control), (b) O-CMC, (c) gallic acid, (d) ascorbic acid, (e) vanillin, (f) O-CMC + gallic acid, (g) O-CMC + ascorbic acid, and (h) O-CMC + vanillin (1:0.5 (*w*/*v*)). Fresh-cut strawberry halves were immersed in the prepared test solutions for 5 min, and then air-dried under sterile airflow in a biological hood for 1 h. After drying, the pieces were randomly distributed among polyethylene terephthalate (PET) containers, ten pieces per container, and stored for up to 7 days at 4 °C, with sampling and quality evaluation after 2, 5, and 7 days of storage. 

### 2.4. Fruit Quality Characterization 

#### 2.4.1. Firmness Test

Firmness was evaluated with an LT-Lutron FG-20KG electronic penetrometer force gauge (Lutron Electronic Enterprise, Indonesia) with an 11 mm probe. During the test, the probe was inserted at two points on the equatorial line of each fruit (20 measurements/treatment). Mean values of five fresh-cut fruit were considered for each treatment. 

#### 2.4.2. Weight Loss Test

Weight loss was calculated by weighing the fresh-cut fruit before and after storage. Results were accounted using the means of five cuts and presented as percentage weight loss. The weight loss percentage was calculated by the following equation:Weight loss (%) = {100 − [(Avg W_S_ ×100)/Avg Wt_0_]}

Avg Wt_0:_ the average weight of the specifically designated fruit cuts at time T0. 

Avg Ws: the average weight of the same designated fruit cuts on the specific sampling day (2, 5, and 7 days).

#### 2.4.3. Microbiological Analysis

The fruit were sampled for microbiological analyses after 2, 5, and 7 days of storage. Samples [approx. 3.5 g] were immersed in 15 mL sterile saline solution (0.9% NaCl) in centrifuge tubes and vigorously vortexed for 2 min. Mold and yeast counts were observed by surface inoculation on a petri dish containing potato dextrose agar (PDA) medium supplemented with 100 ppm chloramphenicol to control bacterial growth (PDA + A). The total microbial count was determined by surface inoculation on plate count agar PCA. After incubation for 48 h at 30 °C, the number of colony-forming units (CFU) was counted and expressed as CFU/g fruit. The reported values are averages of five replicates. 

#### 2.4.4. Color Evaluation

Changes in the color of the strawberry were analyzed daily by measuring the CIELAB parameters L* (lightness), a* (green to red axis), and b* (yellow to blue axis) using a CR-400 Chroma Meter (Konica Minolta Sensing). Hue angle calculations were performed by the following formula [32]:Hue angle = [Arc tan (b*/a*)] (deg)

The results were shown as the mean values of three replicates.

#### 2.4.5. Incidence of Fruit Decay

The decay incidence was determined by calculating the percentage of infected pieces out of the total number of fruit halves per replication. The results are presented as means of three replications per treatment. 

#### 2.4.6. Total Soluble Solids (TSS) Analysis

TSS were evaluated after the 2nd, 5th, and 7th day of treatment by using a digital refractometer (ATAGO Ltd., PR-1, Japan, Tokyo), and the results were presented as degrees Brix (°Brix). In each treatment, three fruit halves were squeezed through a two-layered gauze and the juice was used to measure the TSS. Results were shown based on triplicates.

### 2.5. Sensorial Tests

The effect of different treatments on the fruit aroma was tested by a panel of 5 trained assessors after 7 days of storage. The following olfactory descriptors were evaluated: typical strawberry aroma, sour, rotten, added odor of active agent, and generally pleasant aroma on a scale of 1 to 9. 

### 2.6. Statistical Analysis 

The evaluations were performed at least in triplicate. One-way analysis of variance (ANOVA) and Tukey honestly significant difference (HSD) pairwise comparison tests at *p* ≤ 0.05 were applied by means of the JMP statistical software program, version 7 (SAS Institute Inc., Cary, NC, USA).

## 3. Results and Discussion

A gradual weight loss was observed from all the fruit during the test period, whilst the highest weight loss was seen in untreated fruit (control), followed by gallic acid treated strawberries (Figure 1). Fruit treated with only O-CMC and O-CMC ascorbic had shown a decrease in weight loss of around ~2.5 times less than the untreated fruit, and showed a significantly lower weight loss in comparison to the control. The most beneficial treatment in terms of weight loss reduction was O-CMC-vanillin, as even after seven days, this treatment showed minimal weight loss. Interestingly, the addition of O-CMC to the active substances resulted in the reduction of weight loss in comparison to the treatments which contained active agents only. It is a known fact that oligosaccharides lack the film forming capacity which polysaccharides obtain, yet it was shown that they can self-assemble and form supramolecular structures including films in some cases [33]. Furthermore, oligosaccharides perform as elicitors/wound signals inducing a defense response including wound healing, e.g., lignification [34], which might explain their effect on the weight loss reduction. This finding has further increased the value of the produced oligomers, leading us to suspect their bioactivity and properties.

As one of the key parameters of produce freshness is its color, a*, b*, L*, and hue angle color parameters were tested. In all fruits, their original red color faded during storage and became more brown, resulting in the increase in a* values and decrease in b*, L*, and Hue angle parameters. The Hue angle describes the relative amounts of redness and yellowness, and it can be clearly seen that all of the tested parameters were in good correlation. It can also be seen that as a* values increased, the browning process enhanced and the Hue angle decreased, indicating an increase in the red color. Amongst O-CMC-containing formulations, the best color maintenance was demonstrated by fruit treated with O-CMC-vanillin. Generally, all treatments containing O-CMC were found to improve fruit color parameters during the storage period (Figure 2). Additionally, it was found that treatment with gallic acid resulted in a fast and high browning process, due to the increased oxidation of the polyphenol, resulting in the lowest lightness values.

As the previous tests showed, the treatment of strawberries with O-CMC and active ingredients had a significant effect on the color parameters, and in turn, the quality of fruit during storage. To further explore the impact of the coatings on fresh cuts’ quality, the measurement of TSS (which in strawberries is contributed by three primary sugars, namely, glucose, fructose, and sucrose) as °Brix was checked. All strawberries exhibited a loss of TSS content (Table 1) over the storage time from the average value tested in T0 (11.7 °Brix). The highest TSS change was seen in untreated fruit and those treated with gallic acid. The O-CMC treatment significantly improved the TSS values, causing a subtler decrease. Fruit treated with O-CMC and O-CMC-vanillin exhibited the highest TSS content and the lowest decrease during all storage periods. 

Fruit firmness is another important parameter to check the quality of postharvest produce. The untreated fruit and fruit that were treated with gallic acid demonstrated the highest loss of their firmness during the storage period. Fruit treated with O-CMC and O-CMC-vanillin exhibited the best firmness retention (Figure 3), further indicating the beneficial properties of the combination between the oligomers and vanillin.

Additionally, it was found that fruit treated with O-CMC and O-CMC-vanillin had the smallest percentage of rotten fruit in comparison to other treatments. Furthermore, the fruit treated with these formulations looked overall brighter and fresher at all time periods, resulting in good looking fresh-cut fruit even after 7 days of storage (Figure 4 and Appendix A).

All fruits were observed to visually evaluate their decay progress. Decay incidents were evaluated by the appearance of any apparent physical spoilage of the fruit, mainly multiple black small spots (presumed to be due to the contamination of *Colletotrichum acutatum* fungus), big brown spots of decay, etc. It was seen that treatment with O-CMC and O-CMC vanillin prevented up to 80–100% of fruit decay during all time periods (Figure 5 and Appendix A). 

Microbiological tests further showed that the O-CMC-vanillin treatment demonstrated the most effective reduction during all storage periods, when even after 7 days there was an overall bacterial (~2 log CFU/g) and yeast/mold (~3 log) proliferation inhibition in comparison to other formulations and the control (Figure 6). Overall, it is known that gallic acid, ascorbic acid, and vanillin have antimicrobial and antifungal effects [35,36,37]. However, gallic acid was previously shown to express phytotoxicity which may cancel its positive potential as it has a very low effect on the bacterial proliferation on our fresh-cut fruits [38]. Interestingly, in correlation with our previous results, the addition of O-CMC to all formulations improved the treatments’ antimicrobial and antifungal properties. Furthermore, it can be clearly seen that O-CMC on its own possesses such attributes, while the polysaccharide CMC does not. As oligosaccharides are more linear and small molecules, they might reveal the activity which lacks in the big and complex chemical structure of the polysaccharide. In addition, their combination with other active materials allows them to create strong OH bonds resulting in an efficient and active treatment such as O-CMC vanillin. 

Odor is one of the most important parameters in the evaluation of fruit quality by a consumer. Odor evaluation tests were carried out according to five parameters: typical strawberry odor, sour, rotten, active material odor, and overall pleasant aroma (Figure 7). At the end of storage (t7), the rotten odor prevailed in the control and in the treatments including gallic acid. All fruit treated with acids demonstrated a slight sour odor. Interestingly, when acid was combined with the O-CMC, this sour odor effect was diminished. It was found that O-CMC did not affect the typical aroma of strawberries. As vanillin is a known volatile material, it was seen that strawberries treated with vanillin had a strong scent of the active material; however, this odor reduced over the time. The addition of O-CMC to vanillin diminished the vanillin odor over the tested period, resulting in a pleasant fruit aroma, and once again showing the added impact of the combination with O-CMC. 

## 4. Conclusions

Oligosaccharides produced from carboxymethyl cellulose (O-CMC) alone and their combinations with three natural active agents (ascorbic acid, gallic acid, and vanillin) were tested as treatments for the enhancement of quality and storability of fresh-cut strawberries. All the O-CMC-containing treatments resulted in the improvement of fruit quality, color, firmness, bacterial contamination, etc., in comparison with the treatments that contained active agents only and the control. The O-CMCvanillin formulation showed the best results as it maintained improved fruit properties and controlled the microbial growth, preventing bacterial proliferation and yeast/mold growth even after 7 days of storage. In addition, the O-CMCvanillin treatment has also diminished fruit decay and moisture loss during all storage periods. Interestingly, O-CMC has shown new, novel properties in comparison to the original CMC polysaccharide. It was previously found that oligosaccharides have eliciting potential, stimulating defense mechanisms or stress-induced responses in plants, and improving the quality and safety of the yielded food products [39,40]. Several applications of oligosaccharides, such as O-chitosan, O-pectin, O-alginate, and more, were shown to improve plant growth, nitrogen fixation, etc. In addition, they promote plant defense against viruses, bacteria, and fungi. Moreover, O-alginate was found to improve the quality and storability of plants and fruits, in particular of strawberries [20]. All of these findings imply the possible beneficial effects of others such as O-CMC. Although little is known about the effect of O-CMC as a fruit coating, we have shown previously that O-CMC has an elicitation effect on broccoli sprouts, allowing improved yield acceptance, root elongation, etc. [22]. Additionally, based on the tests conducted, we believe that the developed O-CMC-based formulations have potential as safe, bioactive, biodegradable, and cost-effective materials for maintaining the quality of fresh and fresh-cut food products. This finding could contribute to the reduction of the worldwide food loss problem.

## Figures and Tables

**Figure 1 foods-11-01117-f001:**
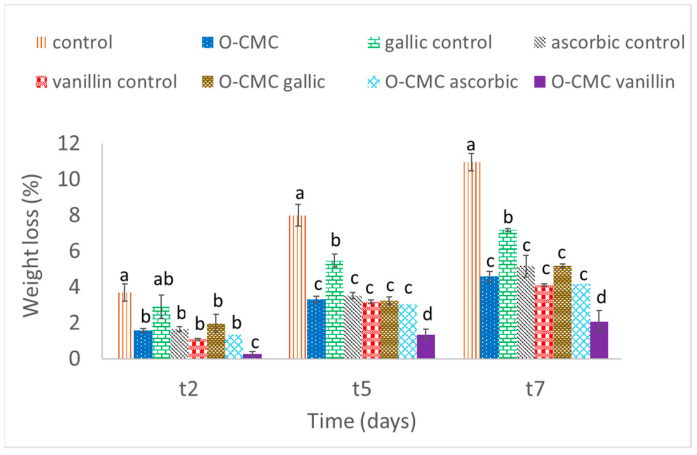
Weight loss (%) of fresh-cut strawberries treated with oligo-CMC formulations after 2, 5, and 7 days of storage at 4 °C (t2, t5, and t7, respectively) as the means of 5 fruit cuts. Values followed by a different letter within the same sampling time are significantly different according to Tukey HSD test at *p* ≤ 0.05.

**Figure 2 foods-11-01117-f002:**
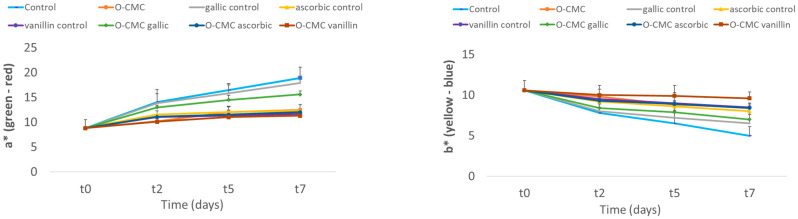
Fruit surface color parameters: a*, b*, L*, and Hue angle at different storage times of fresh-cut strawberries treated with oligo-CMC formulations as the means of three replicates.

**Figure 3 foods-11-01117-f003:**
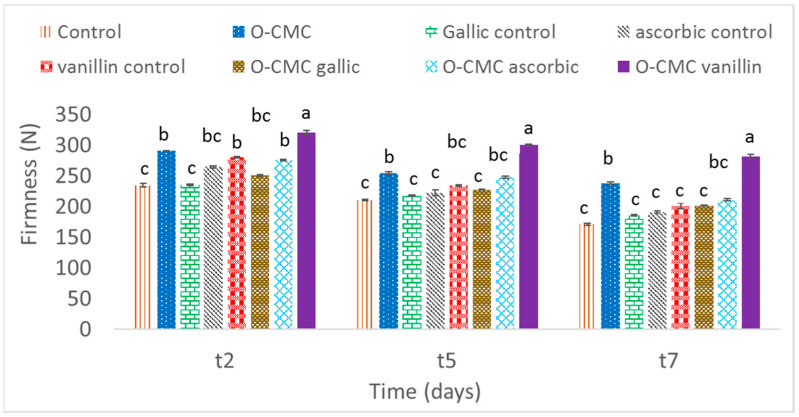
Firmness of fresh strawberries treated with oligo-CMC formulations at three testing times after 2, 5, and 7 days (t2, t5, and t7, respectively) as the means of 5 replicates. Values followed by a different letter within the same sampling time are significantly different according to Tukey HSD test at *p* ≤ 0.05.

**Figure 4 foods-11-01117-f004:**
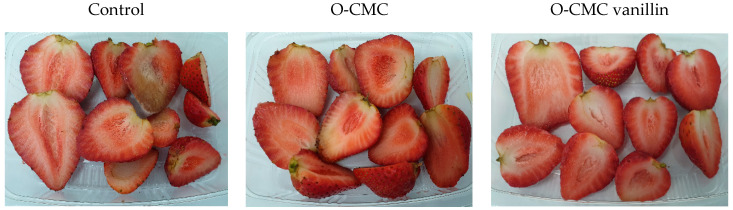
(Left to right) untreated fresh-cut strawberries (control); treated with O-CMC and O-CMC vanillin formulations after 7 days of storage at 4 °C.

**Figure 5 foods-11-01117-f005:**
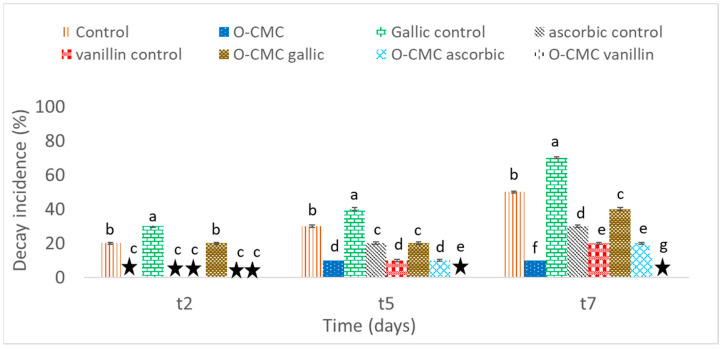
Decay incidence (%) on the strawberries after 2, 5, and 7 days of storage (t2, t5, and t7, respectively). The sign 
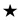
 designates cases when no decay was detected. Values followed by a different letter within the same sampling time are significantly different according to Tukey HSD test at *p* ≤ 0.05.

**Figure 6 foods-11-01117-f006:**
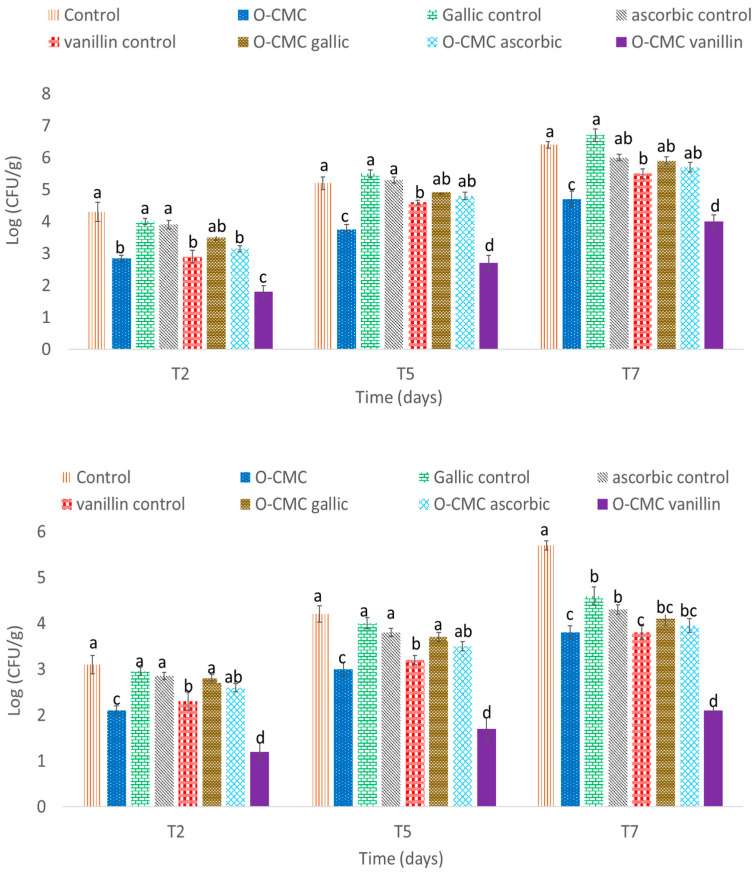
(**up**) Total microbial count on PCA plates and (**down**) mold/yeast count on PDA + A plates of fresh-cut melon after 2nd, 5th, and 7th day of storage at 4 °C treated with oligo-CMC formulations. Values marked by a different letter within the same sampling time are significantly different according to Tukey HSD test at *p* ≤ 0.05.

**Figure 7 foods-11-01117-f007:**
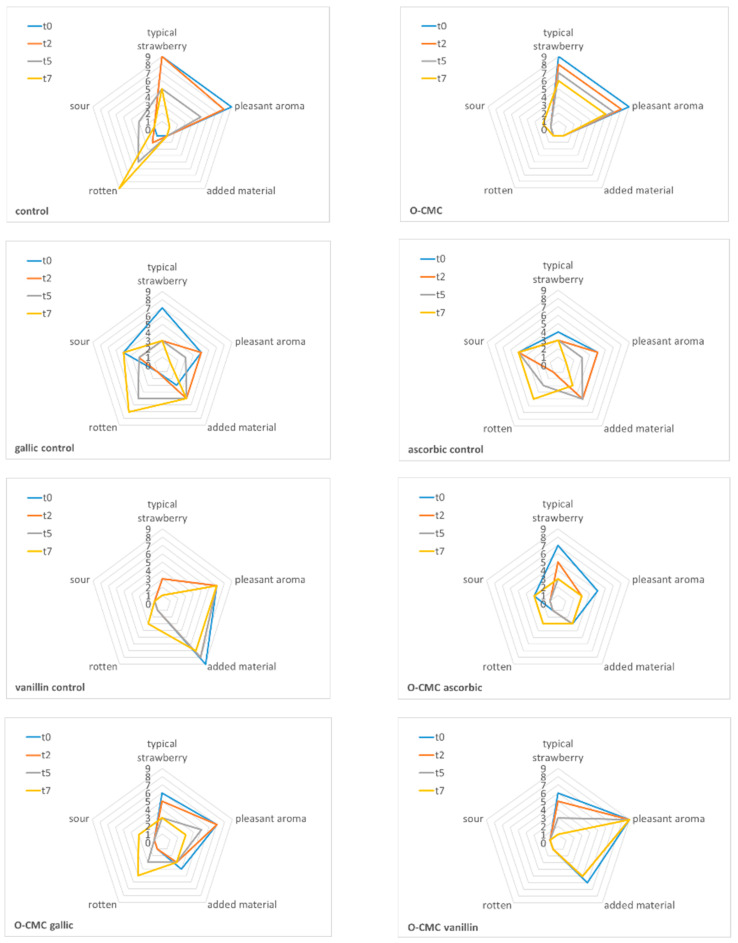
The effect of different treatments on the fruit aroma after 0, 2, 5, and 7 days of storage. The following olfactory descriptors were evaluated: typical strawberry, sour, rotten, added odor of active agent, and generally pleasant aroma on a scale of 1 to 9.

**Table 1 foods-11-01117-t001:** TSS values of fresh-cut strawberries with oligo-CMC formulations at three testing times after 2, 5, and 7 days (t2, t5, and t7, respectively) as the means of three replicates. Values followed by a different letter in each tested day (a–c, a’–c’, and a”–c”) within the same sampling time are significantly different according to Tukey HSD test at *p* ≤ 0.05.

Treatment	TSS (°Brix)		
	T2	T5	T7
control	9.8 ± 0.73a	8.1 ± 1.2b′	7.3 ± 1.1c″
O-CMC	11.1 ± 1.3a	10.7 ± 0.4 a′	10.2 ± 0.65a″
Gallic control	9.9 ± 1.6a	8.6 ± 1.2b′	7.5 ± 1.3c″
Ascorbic control	10.2 ± 1.3a	9.4 ± 0.9b′	8.8 ± 0.5b″
Vanillin control	10.8 ± 1a	10.3 ± 0.3a′	9.9 ± 0.8a″
O-CMC gallic	10.1 ± 1.1a	8.9 ± 0.8b′	7.9 ± 0.6b″c″
O-CMC ascorbic	10.6 ± 1.5a	10.1 ± 1.6a′	9.3 ± 0.7a″b″
O-CMC vanillin	11.5 ± 0.7a	11.3 ± 1.2a′	10.9 ± 0.7a″

## Data Availability

Data supporting the results of this study are included in the present article.

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
