# Peer review of "Oligomers of Carboxymethyl Cellulose for Postharvest Treatment of Fresh Produce: The Effect on Fresh-Cut Strawberry in Combination with Natural Active Agents"

_foods, 2022, doi:10.3390/foods11081117_

Round 1

Reviewer 1 Report

In this work, O-CMC alone and its combinations with three natural active agents (ascorbic acid, gallic acid and vanillin) were tested as treatments for enhancement of quality and storability of fresh-cut strawberries. Although the experimental results are carefully demonstrated, the results lack of analysis and explanation. Therefore, it seems to be an experimental report than a research paper. Suggestions are as follows:

  1. The novelty of this work should be clearly addressed in introduction.
  2. The discussions should be solid.

Reviewer 2 Report

The manuscript is about the use of oligomers of carboxymethyl cellulose (O-CMC) alone or in combination with natural active agents (ascorbic acid, gallic acid and vanillin) as protective edible coatings of strawberries. I consider the manuscript with scientific utility.    

Observation:
- Add a reference of literature to the formula used in the experiment (Materials and Methods, line 124) or some discussion about its meaning in Results and Discussions. 

Suggestion:
- In the caption of Figure 2 add the meaning of parameters: a, b and L
e.g. a* (green to red axis), b* (yellow to blue axis) and L* (lightness).
- Add a discussion about the biochemical mechanism of role of oligomers of carboxymethyl cellulose as a protective edible coating, if known. 

Minor corrections: 
- Use 50 mL rather than 50 ml (L in uppercase) (lines 68, 88, 91)
- Add the point of abbreviation in the name of Journals of references: 14 and 25 - e.g. J. Mater. Chem. B. rather than J Mater Chem B. (Reference 25).  

Reviewer 3 Report

Authors reviewed the potential use of oligomers of carboxymethyl cellulose in combination with natural active agents for postharvest treatment of fresh produce.

Introduction:

L55-56 you mentioned this statement “the development of safe, green, water soluble, cost effective formulation on the basis of O-CMC is of great importance”

Do you think the use of O-CMC is safe and green? We know that carboxymethylcellulose (CMC) are synthetic. Consumers are more concerned about good eating habits and the harmful effects of synthetic/chemical additives in food products. They want to consume healthier and more natural foods. Please refer to the following article :

Benoit Chassaing, Charlene Compher, Brittaney Bonhomme, Qing Liu, Yuan Tian, William Walters, Lisa Nessel, Clara Delaroque, Fuhua Hao, Victoria Gershuni, Lillian Chau, Josephine Ni, Meenakshi Bewtra, Lindsey Albenberg, Alexis Bretin, Liam McKeever, Ruth E. Ley, Andrew D. Patterson, Gary D. Wu, Andrew T. Gewirtz, James D. Lewis, Randomized Controlled-Feeding Study of Dietary Emulsifier Carboxymethylcellulose Reveals Detrimental Impacts on the Gut Microbiota and Metabolome, Gastroenterology, 162(3), 2022, 743-756, https://doi.org/10.1053/j.gastro.2021.11.006.

Please add a paragraph related to the safety issues of CMC in food applications. Many studies reported that CMC promote development of diseases associated with microbiota dysbiosis.

What about O-CMC in terms of safety?

Results & Discussion

You have just exposed the results; please discuss the physical-biochemical phenomena related to the changes of color, texture...etc. Further discussion of the results is needed

Conclusion:

Authors should include their own critical thoughts, what are the major barriers, opportunities, obstacles for O-CMC applications for post-harvest treatment of fresh produce.

Reviewer 4 Report

Technically, the paper has merit. However, some major revisions are required.

General observations/questions

Abstract: More specific results should be included in the abstract

Lines 39 – 42: The authors mentioned the use of some natural antimicrobial agents and edible coatings, it would be good to discuss a few.

Lines 47 – 48: “Interestingly, studies of carboxymethyl cellulose-based oligomers (O-CMC) are generally scarce”. Are there any? Mention the few available.

The introduction should be rewritten to include some relevant work.

Elaborate more on the distribution of the strawberries for the test – Line 83

How the weight loss was calculated (equation) should be included in the methods.

More details on how the incidence of fruit decay was determined should be included.

The unit of TSS should be changed to °Brix or Degrees Brix.

The results and discussion for all the evaluated parameters were combined making the read a bit difficult. The authors should separate this.

“The most beneficial treatment in terms of weight loss reduction was O-CMC-vanillin”. Any reason for this observation?

Quality graphs with more distinct lines should be used in Figure 2. Also, they should be combined.

Table 1 should be redrawn. An explanation for a', a'' etc. should be given.

Figure 3: Correct the caption (remove “A”).

The star symbol in Figure 5 should be explained.

Figure 6 caption - the description mentioned left and right but this isn’t the case

Generally, the in-depth discussion of results obtained is lacking, Authors should improve the discussion.

Based on the above remarks, I would recommend a MAJOR REVISION.

Round 2

Reviewer 1 Report

The revision has been improved and suggest the acceptance.

Author Response

We thank the reviewer for his comments and suggestion to accept the article for publication.

We further checked English language spelling to be on the safe side.